# Combined Trifocal and Microsurgical Testicular Sperm Extraction Enhances Sperm Retrieval Rate in Low-Chance Retrieval Non-Obstructive Azoospermia

**DOI:** 10.3390/jcm11144058

**Published:** 2022-07-13

**Authors:** Marco Falcone, Luca Boeri, Massimiliano Timpano, Lorenzo Cirigliano, Mirko Preto, Giorgio I. Russo, Federica Peretti, Ilaria Ferro, Natalia Plamadeala, Paolo Gontero

**Affiliations:** 1Department of Urology, Urology Clinic—A.O.U. “Città della Salute e della Scienza”, Molinette Hospital, University of Turin, 10100 Turin, Italy; massimiliano.timpano@unito.it (M.T.); dr.lorenzocirigliano@gmail.com (L.C.); mirko.preto@unito.it (M.P.); federica.peretti@unito.it (F.P.); ilaria.ferro@gmail.com (I.F.); natalia.plamadeala@unito.it (N.P.); paolo.gontero@unito.it (P.G.); 2Department of Urology, Foundation IRCCS Ca’ Granda—Ospedale Maggiore Policlinico, University of Milan, 20122 Milan, Italy; dr.lucaboeri@gmail.com; 3Department of Urology, University of Catania, 95123 Catania, Italy; giorgioivan1987@gmail.com

**Keywords:** non-obstructive azoospermia (NOA), sperm retrieval, trifocal testicular sperm extraction (TESE), microsurgical testicular sperm extraction (M-TeSE)

## Abstract

Background: Low-chance retrieval non-obstructive azoospermic (NOA) patients are a subpopulation of NOA patients. The objective of this study is to compare the surgical outcome of microsurgical-assisted testicular sperm extraction (M-TeSE) and combined trifocal/M-TeSE in low-chance retrieval NOA patients. Material and Methods: A single-center retrospective analysis of NOA patients who underwent testicular sperm extraction was performed. Low-chance retrieval NOA (testicular volume < 10 cc and FSH > 12.4 UI/L) was set as the inclusion criteria. Re-do TeSE procedures were excluded from the current analysis. Data were extrapolated from clinical records and operative notes. We compared data from patients who underwent classic M-TeSE (group A) with that from patients submitted to combined trifocal/M-TeSE (group B). Sperm retrieval rate (SRr) was the primary outcome of the study. Surgical outcomes and postoperative complications were evaluated. A multivariate analysis was conducted to investigate predictive factors for positive SR. Results: Overall, 80 patients (60 patients in Group A and 20 patients in Group B) fulfilled the inclusion criteria. The average (SD) age was 35 (8.2) years. The average preoperative FSH was 27.5 (13) UI/L. The average testicular volume was 6.3 (3) cc on the left side and 6.8 (2.5) cc on the right. Groups were similar in terms of preoperative parameters. The overall SRr was 28%. Patients in group B had higher SRr than those in group A (29.4% vs. 26.9%, *p* < 0.03). We identified a significant association between testicular histopathology and positive SR (hypospermatogenesis 100%, spermatogenic arrest 32%, and Sertoli cell-only syndrome 22%). The histopathology report was the only significant predicting factor for SR in the multivariate analysis. Conclusion: The combined trifocal and M-TeSE approach is safe and may represent a valuable approach to enhance the SRr in low-chance retrieval NOA. The histopathology report is confirmed to be the only valuable predicting factor for a positive SR.

## 1. Introduction

In the general population, up to 15% of couples suffer from infertility [1,2]. In half of these couples, a predominantly male infertility factor is detected [3,4,5]. Among male infertility cases, up to 20% are due to azoospermia [6]. Azoospermia can be classified as obstructive (OA) or non-obstructive (NOA). Overall, NOA is sharply more common, representing up to 60% of cases. NOA still represents a challenge for andrologists, being in most cases secondary to an idiopathic spermatogenesis process failure [6,7]. In 1993, the first successful intracytoplasmic sperm injection (ICSI) procedure using sperm retrieval directly from the testis by conventional testicular sperm extraction (TESE) was performed [8,9]. It is widely known that most NOA is characterized by a focal distribution of normal spermatogenesis areas among the altered parenchyma. Following these leads, a well-known limit of conventional TESE is the impossibility to clearly identify these areas [10,11]. Therefore, in order to improve sperm retrieval rate (SRr), trifocal TESE and microsurgical-assisted TESE (M-TeSE) have been proposed as alternative surgical options [10,12,13].

However, to date, there is still no definitive consensus on which procedure may represent the best option to be offered to patients with NOA [14]. Among NOA patients, a subpopulation characterized by low testicular volume and high values of FSH may be identified as “low-chance retrieval” [8,15,16,17]. Despite the relatively high incidence of low-chance retrieval men among the NOA cohort, few reports have specifically investigated the TeSE outcomes of this subgroup of patients.

In the present study, we aim to compare the surgical outcomes of M-TeSE and combined trifocal/M-TeSE in low-chance retrieval NOA patients.

## 2. Materials and Methods

### 2.1. Patients

We retrospectively reviewed a single-center database of over 500 azoospermic patients who underwent surgical sperm retrieval procedures. Data were extrapolated from clinical records and operative notes. Patients with a low-chance retrieval NOA that underwent a surgical sperm retrieval attempt between January 2011 and September 2016 were recruited in the current study. All the surgical procedures were performed by three skilled surgeons.

Inclusion criteria were:-Azoospermia confirmed on 2 specimens according to WHO guidelines [18];-Reduced testicular volume (<10 mL);-Increased serum FSH (>12.4 IU/L);-Patients who underwent a M-TeSE or combined trifocal/M-TeSE.

Exclusion criteria were:-Re-do surgical sperm retrieval procedures;-Genetic disorders related to azoospermia;-Varicocele or prior varicocele repair.

Testicular volume was measured using a standardized orchidometer. Patients operated on before February 2014 underwent a standard microsurgical TeSE (group A), whereas the remainders underwent a combined trifocal and microsurgical approach (group B). A comparative cohort analysis between the two different surgical approaches was performed. Sperm retrieval rate (SRr) was set as the primary outcome of the study. Surgical outcomes and postoperative complications were evaluated.

All the procedures included in the study were ruled in accordance with the ethical standards of the declaration of Helsinki and written informed consent was requested from all study participants.

### 2.2. Surgical Technique

M-TeSE was performed under spinal anesthesia. The patient was placed in the supine position. A 2 cm longitudinal incision on the scrotal raphe was performed to gain access to both hemi-scrotums. The dartos muscle and the tunica vaginalis were opened. An ample equatorial incision of the tunica albuginea was fashioned. Through an operating microscope, a direct examination of the testicular parenchyma was performed under magnification (20–25×) (Figure 1). The more dilated, opaque, and white seminiferous tubules, possibly in proximity to the lobular vascular pedicles, were identified. Multiple testicular specimens were sampled. A microsurgical dissection of both superficial and deep testicular lobes was performed with microsurgical forceps in pursuance of preserving as much as possible of the testicular blood supply. A specimen was taken for histopathology examination. The tunica albuginea and vaginalis were reconstructed by running 4–0 polyglactin sutures. The testicle was placed back into the hemi-scrotum. The same procedure was performed on the contralateral testis.

Combined trifocal/M-TeSE was performed in the same aforementioned setting. Initially, the procedure followed the same steps as a standard M-TeSE. Once access was gained to the testicle, three transversal incisions to the tunica albuginea were performed at the upper pole near the head of the epididymis, in the midline, and at the lower pole, respectively (Figure 2). Sharp scissors were then used to excise the exposed parenchyma. The midline incision was then widened and a standard M-TeSE was performed. The same procedure was performed on the contralateral testis.

## 3. Statistical Analysis

Descriptive and inferential data analysis was performed using IBM SPSS Statistics for Windows, version 27 (IBM Corp., Armonk, NY, USA). The normal distribution of variables was tested by the Kolmogorov–Smirnov test. The categorical variables were described using frequency and percentage. Differences between groups were tested by the chi-squared test and the Fisher’s exact test. Continuous variables are presented as the median and interquartile range (IQR) or mean and standard deviation (SD). Differences between groups were assessed by the Student’s independent t-test and the Mann–Whitney U test. Multivariate logistic regression analysis was performed to investigate the association between predictors and SRr. Two-sided *p* < 0.05 was considered statistically significant.

## 4. Results

In total, 80 patients fulfilled the inclusion criteria and were enrolled in the present study. The average age was 35.0 ± 8.2 years. Additionally, 35% of our patients were active smokers and 5% were diabetic type I. The etiology of NOA is summarized in Table 1. The average preoperative FSH was 27.5 ± 13.0 IU/L. The average testicular volume was 6.3 ± 3.0 mL on the left side and 6.8 ± 2.5 mL on the right side. Group A represented the vast majority of patients in our series (75%), whereas the rest of the patients were located in Group B (25%). A bilateral approach was performed in 66.3% of cases. Neither intraoperative nor postoperative significant complications (>grade 1 Clavien-Dindo [19]) were reported.

Testicular atrophy or postoperative hypogonadism were not found at the six-month follow-up, as preoperative and follow-up serum testosterone did not differ (16 (4.5) nmol/L and 15.55 (6.2) nmol/L, respectively; *p* = 0.44). Mean operative time was 103 ± 12 min for Group A and 110 ± 16 min for Group B, without statistically significant differences among groups (*p* = 0.065).

The overall positive SRr was 28%, and was significantly higher in Group B compared to Group A (29.4% vs. 26.9%, *p* < 0.03). The histopathological analysis revealed 86.3% were cases of Sertoli cell-only syndrome (SCOS), 11.3% was spermatogenic arrest (SA) and 2.5% was hypospermatogenesis (HS). The mean Johnsen score was 3.0 ± 1.6. In 12% of patients, tubules with a Johnsen score ≥ 8 were revealed. In a sub-analysis restricted to the patients with a positive SR, the percentage of tubules with Johnsen score ≥ 8 was 35%. Meanwhile, in the patients with a negative SR, the percentage was set at 5%. The average number of sperm vials stored was 4 ± 1.9. A comparison between patients with positive and negative sperm retrieval is shown in Table 2.

We identified a significant association between testicular histopathology and positive SR (HS 100%, SA 32%, and SCOS 22%; *p* < 0.02). Multivariable logistic regression analysis showed that histopathology was the only predictive factor for positive SR (HS: OR 0.43—*p* = 0.004; SA: OR 0.12—*p* = 0.007; SCOS: OR 0.05—*p* = 0.013) after accounting for age, FSH, testicular volume, and Johnsen Score (Table 3).

## 5. Discussion

With the introduction of Assisted Reproductive Technology (ART), particularly of the ICSI, multiple surgical procedures have been proposed to attempt sperm retrieval. Among these techniques, TeSE is the most used worldwide due to well-known satisfactory outcomes. Although TeSE allows an almost 100% success rate in patients suffering from OA [20], SRr is not so high in the NOA population. SRr is even worse in specific populations of NOA patients, such as Klinefelter men [21] and “low-chance” retrieval NOA patients (low testicular volume < 8 mL and high FSH levels > 12.4 UI) [15]. In particular, in low-chance populations, the reported SRrs are as low as 30% [22].

According to current evidence, different surgical approaches have been proposed to attempt SR in NOA patients. To date, we still lack a consensus on which technique is associated with better outcomes.

Testicular sperm aspiration (TeSA) consists of a testicular needle biopsy. The procedure is safe and economic. Nevertheless, it has been demonstrated that TeSE doubles the SRr of TeSA. Therefore, this procedure is not recommended for NOA patients anymore [22].

TeSE represents a milestone in the management of NOA. It was initially described in 1993, when Craft et al. [23] and Schoysman et al. [24] reported the first case of a successful ICSI by direct sperm retrieval through a TeSE. In the preliminary series, up to 50% of patients with NOA successfully retrieved elongated spermatids [8]. Recent evidence has revealed that the absence of a preserved spermatogenesis in a single biopsy cannot exclude the presence of isolated areas in the rest of the parenchyma featured by a normal spermatogenesis. In fact, the histopathological pattern of the testis parenchyma in NOA patients is frequently heterogeneous [25]. Following these leads, a trifocal TeSE was proposed to enhance SRr in NOA patients. Ostad et al. [12] reported that a trifocal TESE was more effective than unifocal TESE for men with NOA. On the other hand, some authors sustain, on behalf of a postoperative hormonal status evaluation, that a trifocal approach to the parenchyma could potentially damage the endocrinological function. This issue has been discredited by a recent meta-analysis showing no significant hypogonadism rate 24 months after the procedure [26].

Following these leads, in 1999, Schlegel introduced the use of the operative optic microscope for a testicular biopsy. The principle for sustaining the application of optical magnification in TeSE was to help the surgeon to select the more dilated, opaque, and white seminiferous tubules [10]. The aforementioned tubules are deemed to have a higher chance of SR. This feature is particularly helpful in NOA patients, due to the heterogenous distribution of the preserved spermatogenesis area. Overall, M-TeSE shows excellent results in terms of SRr, with less complications when compared to conventional TeSE [27,28]. Despite this, there is still controversy on which technique may guarantee better SRr in NOA patients. On one hand, few prospective, not randomized clinical trials have clearly shown that as a microsurgical approach it is superior to the conventional one in terms of sperm retrieval [27,29]. On the other hand, some authors still argue that a conventional approach provides a valuable option to offer to patients, and a recent metanalysis observed no significant differences in SRr between TeSE and M-TeSE in NOA patients [30].

In order to improve the SRr in this peculiar group of patients, Okubo et al. proposed a combined approach (TeSE and M-TeSE) in a small cohort of 17 patients, reporting an overall increase in SRr. These preliminary data were confirmed by Turunc et al. who obtained a SRr of 33.7% in NOA men who underwent TeSE, while SRr increased up to 50.8% when the patients underwent the combined approach [29].

Focusing on the “low-chance retrieval” group of NOA patients, scientific evidence is even scarcer. A single study evaluated SRr obtained through combined trifocal TeSE and M-TeSE [17]. The authors reported that the combination of trifocal TeSE and M-TeSE showed better SRr when compared to a conventional TESE (66.2% vs. 53.8%). Additionally, trifocal TeSE or M-TeSE alone did not prove better than conventional TESE. According to this evidence, the use of a combination technique (trifocal TeSE and M-TeSE) may significantly improve SRr in “low-chance retrieval” NOA patients. This study was limited by the lack of comparison of the SRr of the different techniques in different patient groups, since all the procedures were performed in the same patient. To the best of our knowledge, we report the first study comparing surgical outcomes of M-TeSE and combined trifocal and M-TeSE in low-chance retrieval NOA patients. Our series clearly support the use of combined trifocal and M-TeSE instead of M-TeSE alone (SRr 29.4% vs. 26.9%, *p* < 0.03). This result may be attributed to the sum of benefits granted from both techniques. Indeed, the trifocal technique allows us to explore the testis in three different areas (upper, middle, and lower pole). Taking into account that NOA patients may show scattered foci of spermatogenesis, a more extensive search of the testis could increase the likelihood of catching these foci. On the other hand, the magnification of testicular parenchyma through the microscope may allow the dissection and identification of enlarged seminiferous tubules where some residual sperm production is more likely to be found. Operative time showed no significant differences between the two procedures, probably due to the rapidity of execution of the trifocal TeSE. No intraoperative or postoperative complications were reported in either group despite a more invasive approach. Particularly, the combined techniques did not lead to any case of postoperative hypogonadism.

Due to the low probability of retrieving any functional sperm in NOA patients, the search for any predictive factors for positive SR has gained clinical interest. Many different elements such as the etiology of NOA, testicular volume, hormones levels (testosterone, FSH, LH and inhibin B), genetics, and histopathology have been studied as potential predictors of positive SR. Metabolic fingerprinting has also been proposed for the prediction of sperm retrieval with promising, but still preliminary, results [31]. Some genetic factors have shown important implications: NOA patients with AZFc deletion or Klinefelter’s syndrome have a chance for sperm retrieval, while those chances are extremely low in men with complete AZFa or AZFb deletions [31]. Some studies have assessed pre-operative plasma FSH concentration and testicular volume as a predictor of positive sperm retrieval, but a more comprehensive review of the literature failed to confirm those hypotheses [27,32,33]. In our analysis, age, FSH, and testicular volume, did not prove significant predicting factors for SR. The only parameter that had a significative positive prediction for SR was the histopathology, confirming the current knowledge in the literature [17,34]. A 22% SRr was reported in SCOS, a condition characterized by only normal Sertoli cells filling the seminiferous tubules with very low or absent spermatogenesis [27]. A slightly better result was obtained in SRr with the histological diagnosis of SA, whilst for HS, SR was always obtained as, although characterized by a reduced number of germ cells, all stages of spermatogenesis were present. It is important to notice that, though histopathology demonstrates a positive relation with SR, it does not provide definitive proof as to whether sperm will be found through subsequent TeSE. Moreover, to obtain a compelling testicular specimen would require a previous testicular biopsy, therefore causing more damage to the testis and increasing the risk of further reducing the focal areas of sperm production.

The limitations of this paper are its retrospective nature, as a randomization was not performed, and the asymmetry in the numerosity between the two groups. A prospective randomized controlled study will be needed to confirm our results.

## 6. Conclusions

The combined trifocal and M-TeSE approach is safe and may represent a valuable approach to enhance the SRr in low-chance retrieval NOA. The histopathology report is confirmed to be the only valuable predicting factor for a positive SR.

## Figures and Tables

**Figure 1 jcm-11-04058-f001:**
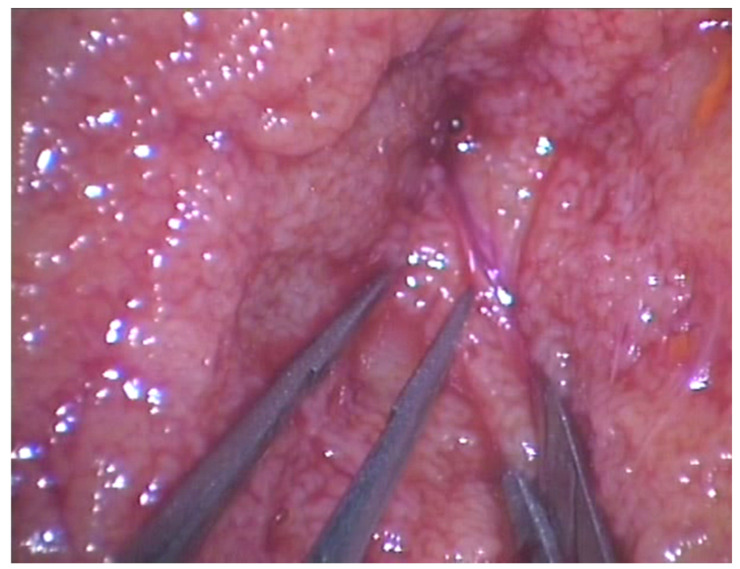
During microsurgical-assisted TeSE, a magnification of 20–25× was obtained through an operative optic microscope in order to identify the more dilated, opaque, and white seminiferous tubules possibly near to blood vessels.

**Figure 2 jcm-11-04058-f002:**
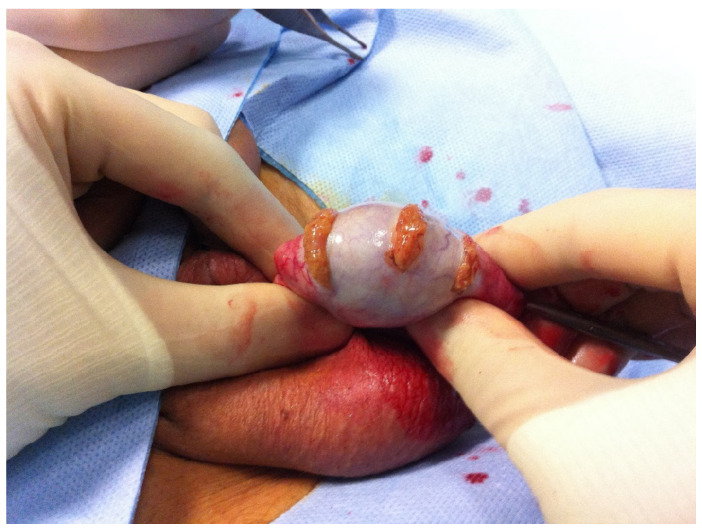
In the trifocal TeSE, the tunica albuginea of the testis is exposed, and three incisions are fashioned at the upper pole, midline, and lower pole.

**Table 1 jcm-11-04058-t001:** Descriptive characteristics for the cohort of 80 patients with low-chance retrieval rate non-obstructive azoospermia.

Figures and Tables		Groups		
Variables	Total	Micro-TeSE	Trifocal Micro-TeSE	*p*-Value
Number of patients, n (%)	80	60 (75)	20 (25)	
Mean age, years (SD)	35 (8.2)	35 (8.7)	38 (7.4)	0.2
Smoking habit, n (%)	26 (32.5)	19 (31.6)	7 (35)	0.08
Mean testicular volume, cc (SD)				
Left side	6.3 (3)	7.2 (4)	6.1 (5)	0.1
Right side	6.8 (2.5)	7.1 (5.7)	6.5 (4.1)	
Mean FSH, UI/L (SD)	27.5 (13)	26 (14.2)	28.2 (13.4)	0.9
Positive sperm retrieval, n (%)	22 (27.5)	16 (26.7)	6 (30)	0.03
Histology, n (%)				
Hypospermatogenesis	2 (2.5)	2 (3.3)	0 (0)	0.07
Spermatogenic arrest	9 (11.2)	7 (11.6)	2 (10)	0.1
Sertoli cell-only syndrome	69 (86.3)	51 (85)	18 (90)	0.09
Mean Johnsen score, n (SD)	3 (1.6)	2.6 (1.7)	3.2 (2)	0.3
Mean sperm vials stored, n (SD)	4 (1.9)	4.5 (1.7)	4 (1.9)	0.4

**Table 2 jcm-11-04058-t002:** Descriptive features between positive and negative SR in the cohort of 80 patients with low-chance retrieval rate NOA.

	Groups		
Variables	Positive SR	Negative SR	*p*-Value
Mean age, years (SD)	35 (6)	35 (9)	0.98
Mean testicular volume, cc (SD)			
Left side	6.2 (3.4)	6.6 (3.1)	0.86
Right side	7.3 (4.1)	7.2 (5.2)	0.09
Mean FSH, UI/L (SD)	24.7 (11.2)	26 (13)	0.74
Histology, n (%)			
Hypospermatogenesis	2 (9,1)	0 (0)	0.73
Spermatogenic arrest	5 (38.5)	4 (6.9)	0.105
Sertoli cell-only syndrome	15 (68.2)	54 (93.1)	0.08
Surgery n (%)			
Micro-TeSE	9 (40.9)	51 (87.9)	0.02
Combined Trifocal + Micro-TeSE	13 (59.1)	7 (12.1)	<0.01

**Table 3 jcm-11-04058-t003:** Multivariate logistic regression analysis testing predictors for positive sperm retrieval.

Variables	Odds Ratio	p Value	[95% Conf. Interval]
Age	1.02	0.64	0.91–1.13
FSH	1.06	0.14	0.98–1.15
Testicular volume	1.15	0.27	0.89–1.5
Hypospermatogenesis	0.43	0.004	0.008–0.5
Spermatogenic arrest	0.12	0.007	0.01–0.25
Sertoli cell-only syndrome	0.05	0.013	0.006–0.12
Johnsen Score	1.59	0.53	0.98–2.58

## Data Availability

Additional data are available from the corresponding author on reasonable request.

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
