# Peer review of "Combined Trifocal and Microsurgical Testicular Sperm Extraction Enhances Sperm Retrieval Rate in Low-Chance Retrieval Non-Obstructive Azoospermia"

_jcm, 2022, doi:10.3390/jcm11144058_

Round 1

Reviewer 1 Report

I belive it is an excellent and interesting study. I wish the authors good luck in future research.

I would like ask the authors to extend the introduction part line 52-57 by studies done by: Untargeted metabolomic profiling of seminal plasma in nonobstructive azoospermia men: A noninvasive detection of spermatogenesis. Kambiz Gilany, Ahmad Mani‐Varnosfaderani, Arash Minai‐Tehrani, Fateme Mirzajani, Alireza Ghassempour, Mohammed Reza Sadeghi, Mehdi Amini, Hassan Rezadoost (Biomedical Chromatography) Metabolic fingerprinting of seminal plasma from non-obstructive azoospermia patients: positive versus negative sperm retrieval. Kambiz Gilany, Naser Jafarzadeh, Ahmad Mani-Varnosfaderani, Arash Minai-Tehrani, Mohammed Reza Sadeghi, Mahsa Darbandi, Sara Darbandi, Mehdi Amini, Babak Arjmand, Zhamak Pahlevanzadeh (Journal of Reproduction & Infertility) In these studies, the authors show that there is more advance method to find spermatogenesis.

Author Response

Thank you for your suggestion, we have added metabolic fingerprint as a possible predictive section in the discussion.

Reviewer 2 Report

This is a retrospective review of mTESE to trifocal mTESE in 80 men. 60 underwent traditional, and 20 trifocal. This is an interesting study and will add to the literature on SSR in NOA> however there are some areas that need to be addressed prior to publication. 

Why was FSH >12.4 and testis volume <10cc l defined as low chance of success? Your reference in the discussion uses testis volume <8cc

Were men with karyotype abnormalities and microdeletions included or excluded?

The authors found and increased SSR of 2.5% for trifocal mTESE, in this sample size of 80 patients, it is hard to believe this reached statistical significance. Statistical review may be required. Additionally, a 2.5% increase is not clinically very important.

Were any patients medically optimized prior to mTESE? if so, how?

Did any patient have a varicocele or prior varicocele repair?

In the discussion, the authors state that SR is lowest in KS, however, this is not correct. Please see https://academic.oup.com/humupd/article/23/3/265/3100961

Was serum testosterone checked before? This should be included, so should post operative T level since testis atrophy and hypogonadism were reported as adverse outcomes. When were post operative hormones checked.

How much sperm was stored per vial? 

Author Response

Author's Reply to the Review Report

Why was FSH >12.4 and testis volume <10cc l defined as low chance of success? Your reference in the discussion uses testis volume <8cc

Thank you for the  advice, we selected the parameters used by Marconi et al. [Marconi, M. et al. Combined trifocal and microsurgical testicular sperm extraction is the best technique for testicular sperm retrieval in ‘low-chance’ nonobstructive azoospermia. Eur. Urol. 62, 713–719 (2012).]. We have added the reference in the inclusion criteria.

Were men with karyotype abnormalities and microdeletions included or excluded?

Did any patient have a varicocele or prior varicocele repair?

Varicocele or prior varicocele repair were considered exclusion criteria. We have updated the exclusion criteria section.

We did not included patients with genetic disorders, we have updated the exclusion criteria section.

The authors found and increased SSR of 2.5% for trifocal mTESE, in this sample size of 80 patients, it is hard to believe this reached statistical significance. Statistical review may be required. Additionally, a 2.5% increase is not clinically very important.

Thank you for the advice. We repeated the statistic and confirmed the results. Although 2.5% is not a high value, it is significant. In the study of Marconi et al. [Marconi, M. et al. Combined trifocal and microsurgical testicular sperm extraction is the best technique for testicular sperm retrieval in ‘low-chance’ nonobstructive azoospermia. Eur. Urol. 62, 713–719 (2012).] the reported difference in SRR is 6%, not so different from the results we achieved.

Were any patients medically optimized prior to mTESE? if so, how?

Thank you for the question. We did not perform previous medical optimization before surgery as  FSH > 12.4 was an inclusion criteria hence medical optimization with FSH or HCG was not indicated.

How much sperm was stored per vial? 

We did not recorded the amount of tissue stored per vial.

In the discussion, the authors state that SR is lowest in KS, however, this is not correct. Please see https://academic.oup.com/humupd/article/23/3/265/3100961

Thank you for the advice. Although the 2017 metanalysis you suggested report a SRR of 50% in KS, we cited a 2020 study ( https://doi-org.bvsp.idm.oclc.org/10.1111/andr.12767 ) that affirms: “In the real-life setting, we observed a lower SRR (21.4%) than that reported in meta-analyses in our cohort of KS patients”

Was serum testosterone checked before? This should be included, so should post operative T level since testis atrophy and hypogonadism were reported as adverse outcomes. When were post operative hormones checked.

Thank you for the question. Testosterone levels were indeed  checked before surgery and 6 months after surgery. We are sorry we did not put the values in the results; we have updated the manuscript with the values.

Reviewer 3 Report

Summary: In this study, microdissection testicular sperm extraction (m-TESE) is compared to trifocal/m-TESE in men with "low chance retrieval" NOA, specifically defined as having a testicular volume < 10 ml and FSH > 12 mIU/ml. The authors found that trifocal/m-TESE had a significantly higher sperm retrieval rate compared with m-TESE alone and that a histological finding of hyperspermatogenesis was predictive of sperm retrieval. 

Overall: There are several major points that need to be addressed and added to the manuscript in order for this data to be usable by reproductive urologists performing sperm retrieval on men with “low chance retrieval NOA.”

Comments:

There are numerous grammatical and punctuation errors that should be addressed.

The figure legend for Figure 2 needs to be corrected (remove the text starting with “Table 1.”)

The described technique for trifocal/m-TESE is unclear in the methods section. In the figure legend for Figure 2, it is clear what the technique for trifocal/m-TESE is. Please change the methods section so that the technique is clear to readers (for example, one incision superiorly, one incision equatorially, and one incision inferiorly).

Please comment on how the patient cohort was selected. One of the primary references in the discussion is to a study that examined a cohort of men with testis size <8ml and FSH >12 mIU/ml. This is an important point to clarify to precisely identify patients for whom this technique may be most useful.

The authors performed a retrospective review. This suggests that patients were selected for trifocal/m-TESE by the surgeon for specific reasons. Although the clinical characteristics of the patients were not significantly different, there could have been some factors that are not listed here for why the surgeon performed trifocal/m-TESE rather than just m-TESE. For example, the surgeon might have thought that trifocal TESE would have been sufficient for retrieving sperm but did not find sperm in these specimens and so a m-TESE was performed - in these cases, sperm retrieval rates may be higher since the surgeon did not think m-TESE was necessary to start with in comparison with the larger cohort where patients all were designated to have m-TESE from the start. Please comment.

Where was the testis biopsy taken for trifocal/m-TESE subjects? Was it done in a similar fashion for both m-TESE and trifocal/m-TESE patients?

How many surgeons performed the operations on the patient cohort? This should be documented in the methods section.

Although the authors state that Table 1 demonstrates the etiology of NOA, there is no mention of etiologies of NOA for the patient cohort (rather only histopathology is shown). Sperm retrieval rates in NOA can range depending on the etiology. Were karyotype and Y chromosome microdeletion studies carried out on these patients? Did any patients have any other risk factors for infertility (ie. history of radiation or chemotherapy or cryptorchidism or orchiectomy)?

Preoperative testosterone levels should be shown in Table 1 to ensure levels were similar preoperatively. Was medical optimization attempted with either group of patients?

It is noted that “postoperative hypogonadism” was not observed in either group. Were serum testosterone levels similar or decreased from baseline a similar amount in each group? This data should be included in Table 1.

How much tissue was removed from patients in each group? This may be hard to estimate, but on average, was there a significantly higher amount of tissue removed from the trifocal/m-TESE group? Can the authors comment on how the findings of higher sperm retrieval rate in the trifocal group is not just due to removal of more tissue (that could be done in the m-TESE alone group as well - if sampling the same areas as the trifocal/m-TESE group)?

In the first paragraph of the results section, the authors state that 66.3% of cases were bilateral. How many of the m-TESE and trifocal/m-TESE cases were bilateral? Was sperm identification performed in the operating room at the time of dissection?

Author Response

Author's Reply to the Review Report

Comments:

There are numerous grammatical and punctuation errors that should be addressed.

Thanks for your advice. We will review the text together with a native English speaker in order to correct any errors.

The figure legend for Figure 2 needs to be corrected (remove the text starting with “Table 1.”)

Thank you. We have updated the manuscript.

The described technique for trifocal/m-TESE is unclear in the methods section. In the figure legend for Figure 2, it is clear what the technique for trifocal/m-TESE is. Please change the methods section so that the technique is clear to readers (for example, one incision superiorly, one incision equatorially, and one incision inferiorly).

Thank you for the advice. We will try to modify the surgical description.

Please comment on how the patient cohort was selected. One of the primary references in the discussion is to a study that examined a cohort of men with testis size <8ml and FSH >12 mIU/ml. This is an important point to clarify to precisely identify patients for whom this technique may be most useful.

We choose the “low-chances retrieval”  NOA patients according to Marconi at al. (reduced testicular volume (< 10 ml); increased serum FSH (> 12.4 IU/L) [Marconi, M. et al. Combined trifocal and microsurgical testicular sperm extraction is the best technique for testicular sperm retrieval in ‘low-chance’ nonobstructive azoospermia. Eur. Urol. 62, 713–719 (2012).]

The authors performed a retrospective review. This suggests that patients were selected for trifocal/m-TESE by the surgeon for specific reasons. Although the clinical characteristics of the patients were not significantly different, there could have been some factors that are not listed here for why the surgeon performed trifocal/m-TESE rather than just m-TESE. For example, the surgeon might have thought that trifocal TESE would have been sufficient for retrieving sperm but did not find sperm in these specimens and so a m-TESE was performed - in these cases, sperm retrieval rates may be higher since the surgeon did not think m-TESE was necessary to start with in comparison with the larger cohort where patients all were designated to have m-TESE from the start. Please comment.

Thank you for the interesting question. We performed a retrospective study starting from 2011. In those years we only performed m-TESE. We started to perform combined trifocal/m-TESE only after the publication of the surgical technique described by Marconi et al. This explains the difference in numerosity between Group A and Group B. The equipe of surgeon believed that we could try to enhance the SRR by adding a trifocal approach to the m-TESE procedure, as reported by Marconi. Years after we decide to evaluate our results. The retrospective nature of this paper is a limitation as a randomization was not performed. Surely a prospective randomized controlled study will be need to confirm our data. I have added this limitation in the paper, thank you.

Where was the testis biopsy taken for trifocal/m-TESE subjects? Was it done in a similar fashion for both m-TESE and trifocal/m-TESE patients?

Thank you for the suitable question. Testis biopsy were randomly taken from the midline incision both in m-TESE that in combined trifocal/m-TESE

How many surgeons performed the operations on the patient cohort? This should be documented in the methods section.

An equipe of three skilled surgeons performed the operations on both groups. I have updated the methods section.

Although the authors state that Table 1 demonstrates the etiology of NOA, there is no mention of etiologies of NOA for the patient cohort (rather only histopathology is shown). Sperm retrieval rates in NOA can range depending on the etiology. Were karyotype and Y chromosome microdeletion studies carried out on these patients? Did any patients have any other risk factors for infertility (ie. history of radiation or chemotherapy or cryptorchidism or orchiectomy)?

We did not considered patients with genetic disorders. As for other risk factors for infertility, we did not gathered those data but in future studies this will be a useful advice, thank you.

Preoperative testosterone levels should be shown in Table 1 to ensure levels were similar preoperatively. Was medical optimization attempted with either group of patients?

It is noted that “postoperative hypogonadism” was not observed in either group. Were serum testosterone levels similar or decreased from baseline a similar amount in each group? This data should be included in Table 1.

Thank you for the question. Testosterone levels were indeed  checked before surgery and 6 months after surgery. We are sorry we did not put the values in the results; we have updated the manuscript with the values.  We did not perform previous medical optimization before surgery as  FSH > 12.4 was an inclusion criteria  hence medical optimization with FSH or HCG was not indicated.

How much tissue was removed from patients in each group? This may be hard to estimate, but on average, was there a significantly higher amount of tissue removed from the trifocal/m-TESE group? Can the authors comment on how the findings of higher sperm retrieval rate in the trifocal group is not just due to removal of more tissue (that could be done in the m-TESE alone group as well - if sampling the same areas as the trifocal/m-TESE group)?

We did not evaluated the amount of tissue removed from patients, but surely combined trifocal/m-TESE imply a higher amount of tissue removed. Nevertheless we believe that the combination of trifocal/m-TESE may increase sperm retrieval not because of a higher amount of tissue, but because of a wider testis exploration (with the others incision in the upper and lower pole) since the distribution of the preserved spermatogenesis area is heterogenous.

In the first paragraph of the results section, the authors state that 66.3% of cases were bilateral. How many of the m-TESE and trifocal/m-TESE cases were bilateral? Was sperm identification performed in the operating room at the time of dissection?

Sperm identification was not performed in the operative room but the biologist took the samples as soon as they were extracted to the laboratory for analysis. There were no significant differences in the amount of bilateral cases between m-TESE and combined trifocal/m-TESE

Round 2

Reviewer 2 Report

Thank you for your edits.